# A searchable image resource of *Drosophila* GAL4 driver expression patterns with single neuron resolution

Geoffrey W Meissner[1]\*, Aljoscha Nern[1]\*, Zachary Dorman[1], Gina M DePasquale[1], Kaitlyn Forster[1], Theresa Gibney[1], Joanna H Hausenfluck[1], Yisheng He[1], Nirmala A Iyer[1], Jennifer Jeter[1], Lauren Johnson[1], Rebecca M Johnston[1], Kelley Lee[1], Brian Melton[1], Brianna Yarbrough[1], Christopher T Zugates[1], Jody Clements[1], Cristian Goina[1], Hideo Otsuna[1], Konrad Rokicki[1], Robert R Svirskas[1], Yoshinori Aso[1]\*, Gwyneth M Card[1]\*, Barry J Dickson[1]\*, Erica Ehrhardt[1], Jens Goldammer[1], Masayoshi Ito[1], Dagmar Kainmueller[2], Wyatt Korff[1]\*, Lisa Mais[2], Ryo Minegishi[1], Shigehiro Namiki[1], Gerald M Rubin[1]\*, Gabriella R Sterne[1], Tanya Wolff[1], Oz Malkesman[1], FlyLight Project Team[1]

[1]Janelia Research Campus, Howard Hughes Medical Institute, Ashburn, United States; [2]Max-Delbrueck-Center for Molecular Medicine in the Helmholtz Association (MDC), Berlin, Germany

**\*For correspondence:**
meissnerg@janelia.hhmi.org (GWM);
nerna@janelia.hhmi.org (AN);
asoy@janelia.hhmi.org (YA);
cardg@janelia.hhmi.org (GMC);
dicksonb@janelia.hhmi.org (BJD);
korffw@janelia.hhmi.org (WK);
rubing@janelia.hhmi.org (GMR)

**Abstract** Precise, repeatable genetic access to specific neurons via GAL4/UAS and related methods is a key advantage of *Drosophila* neuroscience. Neuronal targeting is typically documented using light microscopy of full GAL4 expression patterns, which generally lack the single-cell resolution required for reliable cell type identification. Here, we use stochastic GAL4 labeling with the MultiColor FlpOut approach to generate cellular resolution confocal images at large scale. We are releasing aligned images of 74,000 such adult central nervous systems. An anticipated use of this resource is to bridge the gap between neurons identified by electron or light microscopy. Identifying individual neurons that make up each GAL4 expression pattern improves the prediction of split-GAL4 combinations targeting particular neurons. To this end, we have made the images searchable on the NeuronBridge website. We demonstrate the potential of NeuronBridge to rapidly and effectively identify neuron matches based on morphology across imaging modalities and datasets.

## Editor's evaluation

This study bridges the gap between connectomic data from the fly hemibrain and driver lines needed for functional experiments. The large collection of labeled single cells clones from a large number of samples now provides the community to search both light microscopic and electron microscopic databases for matches using single cells, or cell types. Overall, this manuscript does a compelling job of describing an important resource for the community, which will hopefully be built upon via the collaborative science of many groups as the field develops.

## Introduction

Many experimental approaches to understanding the nervous system require the ability to repeatedly target-specific neurons in order to efficiently explore their anatomy, physiology, gene expression, or function. In *Drosophila melanogaster*, the dominant approaches to targeting cells have been GAL4/UAS and related binary systems (*Brand and Perrimon, 1993*; *Lai and Lee, 2006*; *Pfeiffer et al., 2010*;

*Potter et al., 2010*). The GAL4 protein, expressed from one transgene, binds upstream activation sequence (UAS) elements inserted in a separate transgene and activates the expression and translation of an adjacent functional protein. An extensive toolkit of UAS transgenes has been developed (reviewed in *Guo et al., 2019*). Large collections of GAL4 driver lines have been created, including collections (referred to here as 'Generation 1' or 'Gen1' GAL4 lines) in which GAL4 expression is typically controlled by 2–4 kilobase fragments of enhancer and promoter regions (*Pfeiffer et al., 2008*; *Jenett et al., 2012*; *Tirian and Dickson, 2017*). Published image libraries of the expression patterns of these GAL4 lines are available and provide a basis for visual or computational searches for driver lines with expression in cell populations of interest.

Despite these extensive resources, obtaining precise experimental access to individual neuronal cell types remains challenging. A GAL4 driver line from one of the above collections typically expresses in tens or more neuronal cell types and even more individual neurons, which is not sufficiently specific for many experiments. Several intersectional approaches have been designed to improve targeting specificity (reviewed in *Guo et al., 2019*), the most widely used of which is the split-GAL4 system (*Luan et al., 2006*; *Pfeiffer et al., 2010*). In brief, to create a split-GAL4 driver, the activation domain (AD) and DNA-binding domain (DBD) of GAL4 are individually placed under control of separate enhancer fragments. The AD and DBD are attached to leucine zipper motifs that further stabilize binding. Only in those neurons where both enhancer fragments are active is a functional GAL4 reassembled to activate the UAS, resulting in a positive intersection between enhancer expression patterns. The split-GAL4 system provides the required targeting specificity and has been used at an increasingly large scale (e.g., *Gao et al., 2008*; *Tuthill et al., 2013*; *Aso et al., 2014a*; *Wu et al., 2016*; *Namiki et al., 2018*; *Wolff and Rubin, 2018*; *Dolan et al., 2019*; *Davis et al., 2020*; *Sterne et al., 2021*), but good split combinations remain challenging to predict.

Split-GAL4 construction typically begins with the identification of GAL4 driver lines with expression in the cell type of interest. While the stereotyped shape of fly neurons can sometimes be directly distinguished by visual inspection, the specific features of a neuron are often obscured by other cells in a GAL4 expression pattern. Several stochastic labeling methods that reveal single cells present in broader expression patterns have been developed (reviewed in *Germani et al., 2018*). While large libraries of single-cell images exist (*Chiang et al., 2011*), these were mainly generated using a few widely expressed GAL4 lines. MultiColor FlpOut (MCFO; *Nern et al., 2015*) enables the labeling of stochastic subsets of neurons within a GAL4 or split-GAL4 pattern in multiple colors. In brief, MCFO can use several UAS reporters that are independently stochastically activated by low levels of Flp recombinase. Flp levels can be adjusted to tailor MCFO labeling density for different GAL4 lines or purposes. Labeling a GAL4 pattern using MCFO allows for the efficient determination of a significant fraction of the neurons present within it.

The need for resources to identify single cells of interest using genetic tools (GAL4 lines) has become more urgent due to recent advances in connectomics. Comprehensive electron microscopy (EM) mapping of specific brain regions or whole nervous systems is transforming neuroscience (e.g., *Zheng et al., 2018*; *Maniates-Selvin et al., 2020*; *Scheffer et al., 2020*) by providing anatomy at unparalleled resolution, near complete cell type coverage, and connectivity information. Leveraging these new datasets to understand more than pure anatomy will be greatly facilitated by the ability to genetically target-specific neurons and circuits. Light microscopy (LM) data also complement EM datasets by revealing features outside a reconstructed EM volume or by providing independent validation of cell shapes with a greater sample size. To integrate these formats requires datasets and methods for matching EM neurons with LM-derived GAL4/split-GAL4 data.

Recently developed techniques allow searching for neuron shapes (including neuron fragments, whole neurons, or overlapping groups of neurons) in coregistered LM and EM data. Two leading approaches are NBLAST (*Costa et al., 2016*), which performs comparisons between segmented neurons, and Color Depth Maximum intensity projection (CDM) search (*Otsuna et al., 2018*), which efficiently compares bitmap images using color to represent depth within the samples. NBLAST was recently expanded upon with the combination of PatchPerPix neuron segmentation (*Hirsch et al., 2020*) and PatchPerPixMatch search (PPPM; *Mais et al., 2021*). PPPM identifies neuron segments with similar color and high NBLAST scores that best cover a target neuron of interest, allowing the use of partial segments from densely labeled MCFO samples. Overlapping neurons remain challenging to segment manually or algorithmically, making this an area of rapid development. Advanced

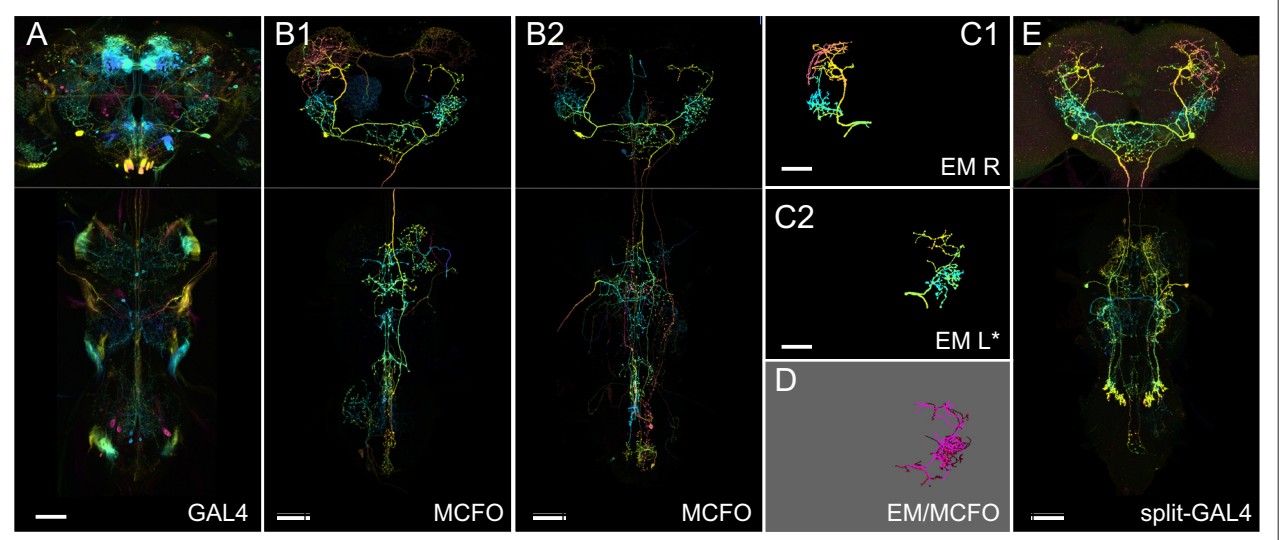

**Figure 1.** Generation 1 MultiColor FlpOut (MCFO) and electron microscopy (EM)/light microscopy (LM) comparison overview. (**A**) Overall GAL4 expression pattern of a driver line containing a cell type of interest, shown as a color depth maximum intensity projection (MIP) (*Otsuna et al., 2018*). Original images are from published datasets (*Jenett et al., 2012*). (**B1**) Example MCFO labeled cells from the driver line in (**A**). MCFO labeling reveals a prominent descending neuron. (**B2**) An additional MCFO labeled cell of the same type but from a different line. The color depth MIPs in B1 and B2 represent data from one of the three MCFO markers, so color changes indicate changes in the *z*-dimension rather than differential MCFO labeling. (**C1, C2**) Matching EM reconstructions for the cell type. Both panels show reconstructions from the right-side Hemibrain; the lower panel is mirrored to facilitate comparison to the LM data. (**D**) PatchPerPixMatch (PPPM) overlay of MCFO from (**B1**) and EM reconstruction from (**C2**). (**E**) Split-GAL4 made from split hemidrivers derived from GAL4 lines in A and B. Driver lines used are R56H09 (**A, B1**), R23E11 (**B2**), and SS01588 (**E**). Hemibrain body IDs are 571346836 (**C1**) and 1786496543 (**C2**). All scale bars, 50 μm.

The online version of this article includes the following figure supplement(s) for figure 1:

**Figure supplement 1.** Generation 1 MultiColor FlpOut (MCFO) expression density categories.

anatomical templates such as JRC2018 improve point-to-point mapping between samples and modalities (*Bogovic et al., 2020*). These search tools and templates bridge the EM/LM gap but require single-cell-level image collections that cover many neurons present within Gen1 GAL4 patterns to reach their maximum utility. In particular, to identify multiple Gen1 GAL4s that can be combined to make a split-GAL4 driver, the morphologies of individual neurons within many GAL4 lines must be available.

Here, we used MCFO to dissect Gen1 GAL4 line patterns at scale to create a resource for linking EM-reconstructed neurons to GAL4 lines, and to improve the process of making split-GAL4 reporters to target neurons, whether they were first identified in EM or LM. We therefore focused on 5155 Gen1 GAL4 lines, most of which have been converted into split-GAL4 hemidrivers, performing three rounds of MCFO labeling to improve coverage of neurons. The resource includes images of 74,337 fly samples, with an average of 14 brain and 7 ventral nerve cord (VNC) images per line. We have released the image data and made it searchable on the NeuronBridge website together with data from the FlyEM Hemibrain and published split-GAL4 lines.

## Results

We used the MCFO approach on Generation 1 GAL4 lines (*Figure 1A*) to visualize individual neurons (*Figure 1B*) making up the GAL4 expression pattern. These neurons can be matched to EM neurons (*Figure 1C, D*) in order to predict split-GAL4 combinations for an EM neuron of interest (*Figure 1E*). We generated two collections of Gen1 MCFO images (*Table 1*). The collection imaged with 20× and 63× microscope objectives targeted particular neurons of interest to collaborators annotating regions primarily in the brain and optic lobes. The collection imaged with 40× objectives broadly canvassed neurons in the central brain and VNC.

**Table 1.** Image collection statistics.

| | 20×/63× collection | 40× collection | Total |
|---|---|---|---|
| Gen1 GAL4 lines | 2463 | 4575 | 5155 |
| Samples | 27,546 | 46,791 | 74,337 |
| Average samples/line | 11.2 | 10.2 | 14.4 |
| Std. Dev. samples/line | 7.6 | 4.6 | 8.7 |
| Average brain/line | 11.2 | 10.1 | 14.3 |
| Average VNC/line | 0.9 | 7.1 | 6.8 |
| Female % | 94.2% | 44.9% | 63.2% |
| 20×/40× image tiles | 29,784 | 111,380 | 141,164 |
| 63× image tiles | 22,775 | – | 22,775 |
| Lines with 63× images | 1748 | – | 1748 |
| Samples with 63× images | 8447 | – | 8447 |

A challenge with any stochastic neuron labeling approach is to optimize the number of identifiable neurons in each sample: too sparse and samples are empty or have few labeled neurons; too dense and the neurons overlap, making it difficult to fully isolate individual neurons even if they are labeled in different colors. MCFO allows for control of labeling density by optimizing the amount of Flp activity, either by selecting different Flp drivers, or altering heat shock duration for hs-Flp (*Nern et al., 2015*). GAL4 lines with broader expression typically require lower Flp activity to yield isolated neurons. In the 20×/63× MCFO collection, labeling density was customized for collaborators focused on annotating particular central nervous system (CNS) regions, iterating on prior results (*Nern et al., 2015*). In the 40× MCFO collection, labeling density was initially standardized (Phase 1), then optimized based on overall GAL4 expression density (Phase 2; *Figure 1—figure supplement 1A*). For many lines, there is no globally ideal level of Flp activity, as they have varying levels of expression density in different CNS regions.

The 20×/63× and 40× datasets differed in several other respects (*Table 1*). The 20×/63× collection was imaged with 20× objectives, followed by 63× imaging of specific regions of interest, whereas the 40× collection was uniformly imaged at 40×. The 20×/63× collection was focused on a smaller set of lines visualized primarily in female brains (94.2%), whereas the 40× collection covered more lines (4575 vs. 2463), a mixture of male and female samples (44.9% female), and both brains and VNCs (7.1 VNCs per line vs. 0.9 in the 20×/63× dataset).

Finally, as the 20×/63× dataset and existing publications (e.g., *Fischbach and Dittrich, 1989*; *Morante and Desplan, 2008*; *Takemura et al., 2013*; *Nern et al., 2015*; *Takemura et al., 2015*) effectively documented the largely repetitive structure of the optic lobes, the 40× dataset excluded them. Collections of split-GAL4 driver lines for many optic lobe cell types are already available (*Tuthill et al., 2013*; *Wu et al., 2016*; *Davis et al., 2020*). Many neurons that connect the optic lobe with the central brain can still be identified in the 40× dataset based on their central brain arborizations. The optic lobe anatomy of such cells could be further characterized in follow-up experiments with the identified GAL4 lines.

## 40× Gen1 MCFO collection

After performing extensive MCFO labeling for the 20×/63× dataset, we performed comprehensive MCFO mapping of Gen1 GAL4 lines across most of the CNS. MCFO labeling of *Drosophila* neurons was performed with a pan-neuronal Flp recombinase (R57C10-Flp) on 4562 Generation 1 GAL4 lines in Phase 1. We generated images of 27,226 central brains and 26,512 VNCs from 27,729 flies. The CNS was typically dissected from six flies per line. A medium-strength Flp transgene (*R57C10-Flp2::PEST in attP18*; *Nern et al., 2015*) was used for almost all lines, yielding a wide range of neuronal labeling in each MCFO sample. 238 of the sparser lines were crossed to an MCFO reporter with a stronger Flp transgene (*R57C10-FlpL in su(Hw)attP8*), and 71 lines were crossed to both reporters.

GAL4 lines were qualitatively categorized into rough groups by density of expression within the central brain and VNC, ranging from Category 1 yielding no unique neurons per sample, to Category 5 being so dense that it overwhelmed our immunohistochemical approach, leaving a shell of partially labeled neurons around the outside of each sample (*Figure 1—figure supplement 1A*). Category 2 lines were characterized by sparse, easily separable neurons, whereas Category 3 yielded denser but identifiable neurons. Category 4 displayed densely labeled neurons that were challenging to distinguish. Most lines ranged between Categories 2 and 4 (*Figure 1—figure supplement 1B*).

In order to increase the number of identifiable neurons, a subset of lines was re-examined with altered parameters. Phase 2 of the 40× pipeline generated images of an additional 18,894 central brains and 6235 VNCs from 19,062 flies. Phase 2 GAL4 expression density was optimized by (1) selecting lines with expression most likely useful for split halves, (2) adjusting MCFO parameters to maximize separable neurons obtained per sample, and (3) limiting brains and VNCs processed per line to minimize the diminishing returns associated with oversampling. Phase 2 focused on Category 2 and 3 lines as most likely to be useful for split-GAL4 creation. Category 1 and 5 lines were outside our effective labeling range and were therefore excluded from further work. High neuron density

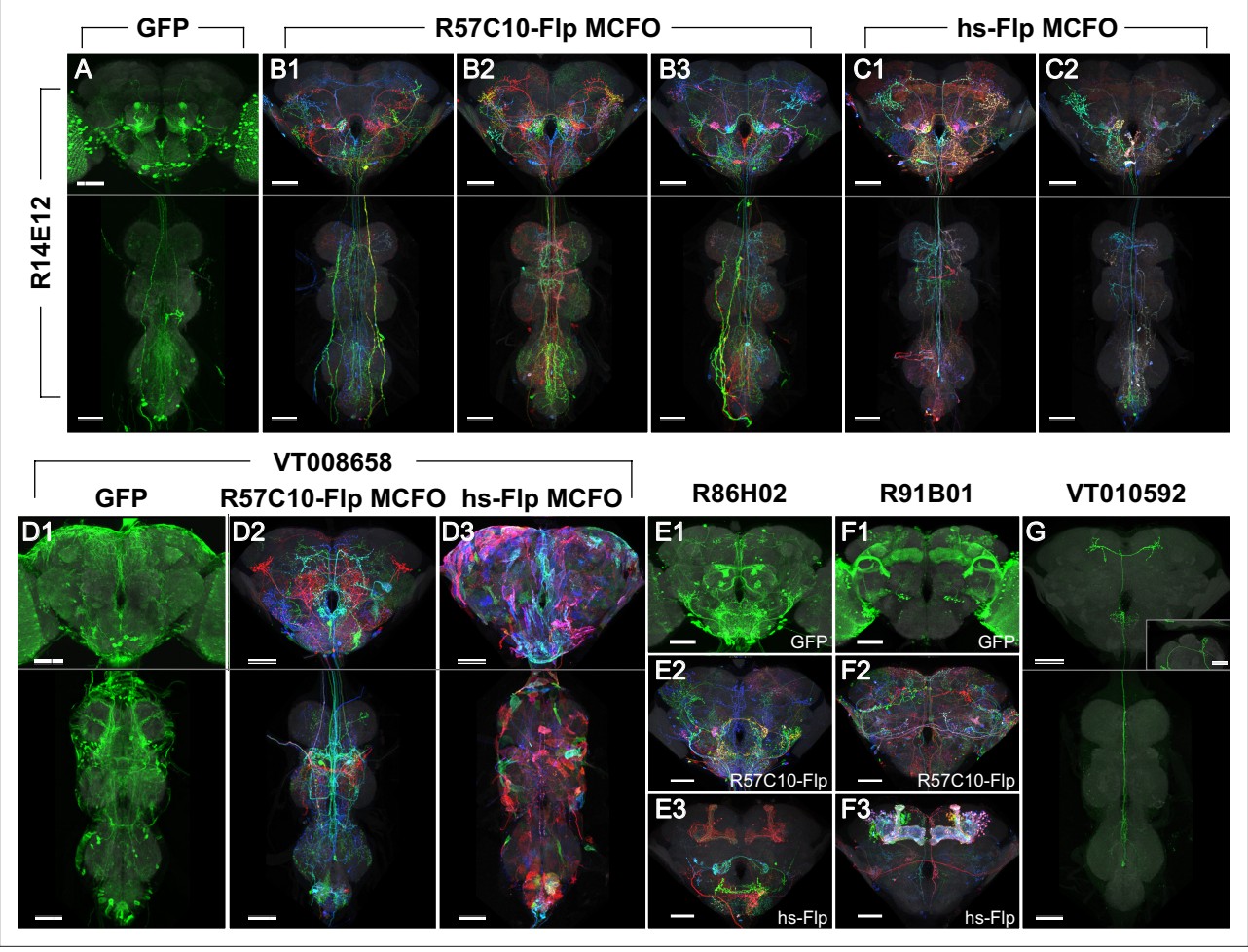

**Figure 2.** Phase 1 and 2 overview and labeling examples. R14E12-GAL4 in attP2 crossed to (**A**) pJFRC2-10XUAS-IVS-mCD8::GFP, (**B**) R57C10-Flp MCFO, or (**C**) hs-Flp MCFO. Adult central nervous system (CNS) maximum intensity projections (MIPs) are shown, with neuropil reference in gray and neuronal signal in green (**A**) or full MultiColor FlpOut (MCFO) colors (**B, C**). Multiple examples are shown for B, C. Scale bars, 50 μm. (**D**) Glia are seen with VT008658-GAL4 in attP2 crossed to (**D1**) pJFRC2-10XUAS-IVS-mCD8::GFP and (**D3**) hs-Flp MCFO, but not (**D2**) R57C10-Flp MCFO. (**E**) Kenyon cell labeling is not seen with R86H02-GAL4 in attP2 crossed to (**E1**) pJFRC2-10XUAS-IVS-mCD8::GFP or (**E2**) R57C10-Flp MCFO, but is seen when crossed to (**E3**) hs-Flp MCFO. (**F**) Kenyon cell labeling is seen with R91B01-GAL4 in attP2 crossed to (**F1**) pJFRC2-10XUAS-IVS-mCD8::GFP and (**F3**) hs-Flp MCFO, but is not seen when crossed to (**F2**) R57C10-Flp MCFO. (**G**) An ascending neuron (sparse T) is commonly seen with many Gen1 GAL4 lines crossed to different reporters. VT010592-GAL4 in attP2 crossed to R57C10-Flp MCFO is shown as an example. A single neuron channel plus reference are shown for clarity. The inset shows a lateral (*y*-axis) MIP of the brain. All scale bars, 50 μm.

within Category 4 means that although the theoretical neuron yield from each sample is high, our ability to distinguish individual neurons is low (although future improvements to neuron segmentation approaches are expected to improve yields).

Heat-shock Flp (hs-Flp) was used in Phase 2 rather than R57C10-Flp (*Figure 2*). While both R57C10-Flp and hs-Flp are theoretically expected to label all neurons, in practice each is likely to have subtle biases as previously proposed (*Nern et al., 2015*; see also below). By switching Flp enhancers in Phase 2, we attempted to mitigate the impact of these biases. The 37°C heat shock duration for hs-Flp was optimized for each density category. Prior results reported by *Nern et al., 2015* indicated that heat shock effectiveness is nonlinear: limited to background activity up to ~10 min, a somewhat linear range between 10 and 20 min, and gradually diminishing returns up to ~40 min; heat shocks longer than an hour begin to harm fly survival. We chose a heat shock duration of 40 min for Category 2 lines to yield as many neurons as possible per sample. For Category 3 a 13 min heat shock provided the desired labeling density similar to Category 3 in Phase 1. To increase the chance of obtaining sex-specific neurons and neuronal morphology, we randomly choose one sex for each half of the lines in Phase 1 and then in Phase 2 switched them to the opposite sex.

As the number of MCFO samples for a given GAL4 line increases, the probability of labeling additional unique neurons diminishes until every neuron labeled by that GAL4 line is represented within the MCFO dataset. Sparser lines approach saturation more rapidly, especially because we can use higher Flp activity to label a greater fraction of available GAL4 neurons per sample without overwhelming detection. Thus, in Phase 2 we processed fewer samples for Category 2 GAL4 lines than for Category 3. In addition to diminishing returns within each GAL4 line, there are diminishing returns within each region of the CNS. Although recent estimates vary (37–100k neurons for the central brain including subesophageal ganglion but not the optic lobes, 15–20k for the VNC; *Bates et al., 2019*; *Godfrey et al., 2021*; *Mu et al., 2021*; *Raji and Potter, 2021*), the adult *Drosophila* central brain has many more neurons than the VNC, suggesting earlier diminishing returns in the VNC. Thus, we focused Phase 2 more heavily on the brain than the VNC, which together with the above density adjustment led to imaging on average 6.0 brains in Category 2 or 9.1 brains in Category 3, and 2.5 VNCs per line across both categories.

## MCFO labeling observations

The large number of lines processed under mostly uniform MCFO conditions provided an opportunity to observe, at scale, some features of MCFO labeling with the specific Flp recombinase drivers used here. Similar observations were noted previously (*Nern et al., 2015*). As with R57C10-GAL4, which contains the same fragment of the *synaptobrevin* enhancer region (*Pfeiffer et al., 2008*), R57C10-Flp is thought to be exclusively expressed in postmitotic neurons. In contrast, hs-Flp is expected to label most if not all cells in the fly, including neurons, glia, and trachea, as reviewed in *Ashburner and Bonner, 1979*. Thus, glial patterns were obtained in 8% of lines (36 of 460 lines tabulated) in Phase 2 with *pBPhsFlp2::PEST in attP3*. This obscured neurons in maximum intensity projections (MIPs), but typically did not impair three-dimensional visualization or searching, and may prove of use for future glial studies (*Figure 2D*). For example, the split-GAL4 approach has also been successfully applied to several types of glia in the optic lobe (*Davis et al., 2020*).

Kenyon cells of the mushroom body were labeled at different rates with each reporter. We scored for the presence of Kenyon cell labeling in a random sample of 10% of the total lines imaged (*n* = 460 lines). Labeling manifested as either distinctly labeled neurons, a relatively faint hazy labeling or both. Kenyon cells were much more commonly labeled using hs-Flp MCFO (430 lines, or 93%) than with R57C10-Flp MCFO (44 lines, or 10%) or UAS-GFP (111 lines, or 24%; *Figure 2*). Most frequently lines had unlabeled Kenyon cells with GFP and R57C10-Flp MCFO and labeled Kenyon cells with hs-Flp (253 lines, or 55%; *Figure 2E*). Lines were also observed with labeled Kenyon cells using GFP and hs-Flp MCFO, but not R57C10-Flp (59 lines, or 13%; *Figure 2F*). As the Kenyon cells are well characterized (and thus an unlikely target for new split-GAL4s), compact, and easily identified, this labeling can be ignored except when substantially brighter than other neurons of interest.

A characteristic ascending neuron (sometimes referred to as 'sparse T') was observed at very high frequency. The neuron(s) has a cell body near the metathoracic ganglion and projections ascending to the anterior then the posterior brain, loosely resembling the letter 'T' in MIP images (*Figure 2G*). It was observed in at least one sample from over 60% of lines crossed to either MCFO reporter (67

lines in Phase 1 and 64 lines in Phase 2, out of 107 lines scored) and was likely present but obscured in other lines. The greater density of labeling in full GAL4 patterns (when crossed to UAS-GFP) made scoring more difficult, yet a similar neuron was seen in 22 of the same 107 lines. This suggests that the high labeling frequency of this neuron in our dataset is a property of the GAL4 collections rather than an artifact of our sampling methods. No other neurons were observed to be so frequently labeled.

## Neuron searching across image collections

This image collection makes it possible to identify GAL4 driver lines with expression in identified single neurons using manual or computational searches without the need for new anatomical experiments. The cellular resolution of the data enables many analyses that are impossible with the existing libraries of full GAL4 driver expression patterns. The single-cell data are particularly useful for identifying a neuron in both EM and LM datasets.

Although LM images do not match the synaptic resolution of EM data, they can provide additional, complementary anatomical information. First, identification of LM matches provides an independent quality check for EM reconstructions (e.g., *Scheffer et al., 2020*; *Phelps et al., 2021*). Second, the LM data often include multiple examples of a cell type and thus provide insights into variable features of cell shapes. Finally, except for the optic lobes, our LM data include the full brain and (for many specimens) VNC and thus provide the full shape of cells that are only partly contained in current EM volumes. For example, the Hemibrain dataset does not fully include neurons that span both brain hemispheres or project to or from the VNC (see *Figure 1*). It is thus important to be able to perform EM/LM matching.

While accurate matching of EM reconstructions with single-cell LM images can sometimes be achieved by direct visual inspection (e.g., *Takemura et al., 2013*), automated approaches for image alignment, segmentation, and search are essential for efficient use of these large datasets. Alignment here was accomplished by registering all LM and EM data to JRC2018 brain and VNC templates (*Bogovic et al., 2020*). We have also made the neuron search tool NeuronBridge (https://neuronbridge.janelia.org/) (*Clements et al., 2022*) publicly available.

NeuronBridge currently allows the user to perform anatomical similarity searches between published datasets reported by Janelia's FlyLight and FlyEM Team Projects. Searching is based on two approaches: (1) Color Depth MIP (CDM), which allows direct comparisons of expression similarity in registered images without the need for a complete skeletonization (*Otsuna et al., 2018*) and (2) PatchPerPixMatch (PPPM), which enhances NBLAST to find groups of neuron segments (identified in our samples by PatchPerPix segmentation) that best match a target neuron (*Costa et al., 2016*; *Hirsch et al., 2020*; *Mais et al., 2021*).

The basic strategy of CDM searching is to represent neuronal expression with a two-dimensional MIP, using color to indicate the third depth dimension. Two aligned brain images can then be compared by looking for pixels of similar color at similar $x$–$y$ coordinates of their color depth MIPs. The color depth MIP search approach used for NeuronBridge was extended in several ways to improve matches for denser MCFO data (*Otsuna et al., 2023*). These include (1) preprocessing the MCFO images with direction selective local thresholding (*Kawase et al., 2015*) 3D segmentation to create a separate color depth MIP for each fully connected component; (2) color depth searching using mirrored EM Hemibrain neurons as masks and MCFO images as target libraries; and (3) weighting of match scores based on signal outside of the search masks.

PPPM searching is based on the evaluation of fully (but often imperfectly) segmented neurons (*Hirsch et al., 2020*; *Mais et al., 2021*). The underlying NBLAST algorithm compares the similarity in 3D location and neuronal arbor orientation at many points along two neuron segments. PPPM looks for an optimal combination of neuron segments that together maximize an NBLAST-derived similarity score for the target neuron. It includes optimizations for identifying non-overlapping segments that tile a target, along with positive weighting for segments of similar color, as would be expected from an MCFO neuron broken into multiple segments.

These comparisons are currently pre-computed as data are added or updated in NeuronBridge, so searching is fast. Searches can begin at NeuronBridge given a GAL4 line name or EM body ID, or from FlyEM's neuPrint (https://neuprint.janelia.org/) (*Clements et al., 2020*; *Scheffer et al., 2020*) and FlyLight's Gen1 MCFO (https://gen1mcfo.janelia.org/) and Split-GAL4 anatomy (https://splitgal4.janelia.org/) websites, leading directly to potential matches in the complementary modality. Search

results are sorted by match quality and displayed for easy comparison (*Clements et al., 2022*). The color depth MIP format is also well suited for fast visual inspection of search results, simplifying the exclusion of false positives, which are difficult to avoid without compromising search sensitivity. Search results are linked directly to corresponding data in other online resources such as Virtual Fly Brain (*Milyaev et al., 2012*).

In addition to pre-computed search results for published datasets, we have also made custom search capability available in NeuronBridge (*Clements et al., 2022*). An unaligned 3D image stack can be uploaded, and the service will register it to the JRC2018 standard reference template (*Bogovic et al., 2020*). CDMs are automatically generated from the aligned image, and an interactive selection tool allows the user to choose a channel and mask a target neuron for the search. Targets can be searched against either the EM or LM image database, in a highly parallel (~3000 threads) cloud-based implementation that completes within a few minutes. Custom search results are browsed in the same way as pre-computed results.

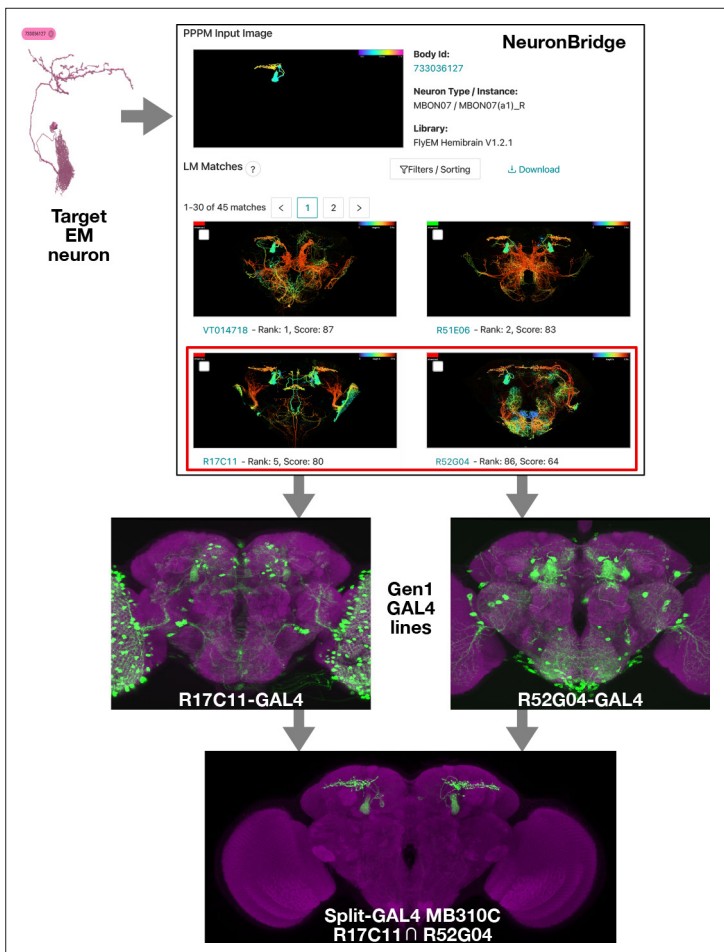

**Figure 3.** Electron microscopy (EM)/light microscopy (LM) search for split-GAL4 creation. Neuron search techniques allow for the identification of Gen1 MultiColor FlpOut (MCFO) images containing an EM body of interest. The corresponding Gen1 GAL4 lines should label the same neuron with other upstream activation sequence (UAS) reporters, as should split-GAL4 hemidrivers constructed with the same enhancer fragment. The two hemidrivers can then be combined into a split-GAL4 with the aim of generating a driver that specifically targets that neuron. An example is shown of the anticipated search process, from a neuron identified via EM to the creation of a split-GAL4 driver. As in *Figure 1*, NeuronBridge displays color depth maximum intensity projections (MIPs) of single MCFO markers rather than the full MCFO image, so color changes indicate depth rather than different neurons. NeuronBridge result order was reformatted for display purposes. The example shown includes FlyEM Hemibrain body ID 733036127 (*Scheffer et al., 2020*), Generation 1 GAL4 lines R17C11-GAL4, R52G04-GAL4, and split-GAL4 MB310C (MBON07) (*Jenett et al., 2012*; *Aso et al., 2014b*).

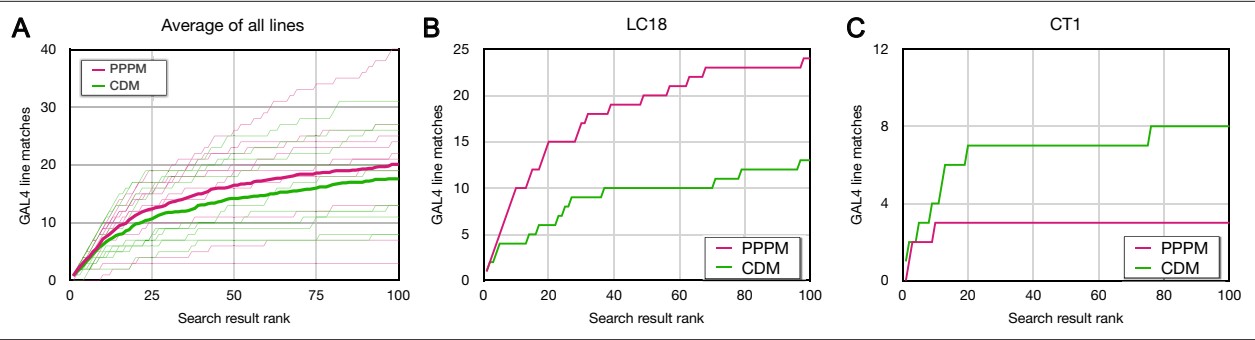

**Figure 4.** Forward analysis: direct evaluation of Color Depth Maximum intensity projection (CDM) and PatchPerPixMatch (PPPM) search results. EM bodies were searched for in Phase 1 40× Gen1 MultiColor FlpOut (MCFO) light images using CDM and PPPM approaches. Search results were qualitatively evaluated by an anatomical expert for the presence of the sought neuron. Most results were scored based on color depth maximum intensity projection (MIP) images. Full image stacks were used to score about 20% of samples, including the majority of samples scored as containing the sought neuron. The cumulative number of correct matches found is plotted against the depth of searching for CDM (green) and PPPM (magenta). (**A**) Average results for each search approach are plotted in bold on top of individual results. (**B**) Cell type LC18 (Hemibrain body 1722342048) search result evaluation. (**C**) Cell type CT1 (Hemibrain body 1311993208) search result evaluation.

The online version of this article includes the following source data and figure supplement(s) for figure 4:

**Source data 1.** Table of forward analysis results by cell type.

**Figure supplement 1.** Forward analysis individual plots for Color Depth Maximum intensity projection (CDM) and PatchPerPixMatch (PPPM).

## Search approach evaluation

We performed limited evaluations of CDM and PPPM search performance between the EM Hemibrain (*Scheffer et al., 2020*) and the Gen1 MCFO dataset in the context of making split-GAL4 lines specifically targeting EM bodies of interest (*Figure 3*).

Search performance can be evaluated in several ways depending on the application (*Costa et al., 2016*; *Otsuna et al., 2018*; *Mais et al., 2021*). We refer here to 'forward' and 'reverse' analysis in the context of split-GAL4 creation. Forward analysis consisted of direct qualitative evaluation of EM to LM search results, determining whether top LM results appeared to contain the searched for EM body. Forward analysis is best performed with detailed knowledge of the examined neurons to avoid false positives, and we restricted our analyzed set of neurons accordingly. Reverse analysis made use of previously documented associations between split-GAL4 lines and EM bodies. If a split-GAL4 line labels a neuron, its constituent split hemidrivers should as well, as should some MCFO of Gen1 GAL4 lines with the same enhancers. We thus evaluated whether known EM/LM matches were highly ranked within the search results. Due to the stochastic nature of MCFO, not every sample of a valid matching GAL4 line will contain the target neuron.

Evaluation of the search approaches also addressed neuron coverage of the Gen1 MCFO dataset. For both search directions the total number of correct matching samples and GAL4 lines gave a measure of how completely the Gen1 MCFO dataset labels each queried neuron.

We performed forward analysis on the top 100 CDM and PPPM Phase 1 Gen1 MCFO search results for 10 Hemibrain bodies (*Figure 4*). Both CDM and PPPM correctly identified many highly ranked matches in the dataset for each examined EM body. CDM identified 17.6 ± 8.3 (average ± standard deviation) correct lines per Hemibrain body, whereas PPPM identified 20.1 ± 10.6.

For cell type LC18, PPPM outperformed CDM, with 24 and 13 correct matches in the top 100, respectively (*Figure 4B*). For cell type CT1, on the other hand, CDM correctly found 8 results in the top 100, compared to 3 for PPPM (*Figure 4C*). More generally, CDM and PPPM each identified many lines in the top 100 results that were not identified by the other search approach (*Figure 4—source data 1*). CDM uniquely identified 8.2 ± 6.1 and PPPM uniquely identified 10.7 ± 8.7 lines, respectively.

Thus, at least for this limited set of neurons, the Gen1 MCFO collection isolates enough examples of each neuron to likely create a split-GAL4 combination. CDM and PPPM successfully identify these correct matches, although they are interspersed with a larger number of false matches. Both approaches varied widely by neuron, without obvious correlation to neuron morphology (*Figure 4—figure supplement 1*). Although all 10 neurons examined here yielded at least 9 matching lines, we

**Table 2.** Reverse analysis: scoring known match search ranks in Color Depth Maximum intensity projection (CDM) and PatchPerPixMatch (PPPM) results.

PPPM and CDM search results on nine Hemibrain bodies were scored for the presence of known GAL4 matches from the literature (*Schretter et al., 2020*; *Wang et al., 2020a*). Only the top-ranking sample for each line and EM body comparison was considered. Searches were performed across only Phase 140× Gen1 MCFO collection data. vpoDN PPPM median line ranks were 7, 49, >400, and >400. Results for bodies 514850616 and 5813063587 are reformatted from *Mais et al., 2021* Figure 9.

| Name | Hemibrain body ID | Known-matching lines | PPPM median line rank | CDM median line rank | PPPM % in top 50 | CDM % in top 50 |
|------|-------------------|---------------------|----------------------|---------------------|------------------|-----------------|
| pC1e | 514850616 | 13 | 14 | 58 | 69% | 46% |
| pC1d | 5813063587 | 12 | 28 | 41 | 58% | 50% |
| aIPg | 645456880 | 5 | 6 | 3 | 80% | 100% |
| oviDN | 550655668 | 4 | 70 | 42 | 50% | 75% |
| oviDN | 519949044 | 4 | 95 | 41 | 50% | 50% |
| SAG | 517587356 | 2 | 49 | 78 | 50% | 0% |
| SAG | 5812981862 | 2 | 44 | 118 | 50% | 0% |
| vpoDN | 5813057864 | 4 | NA | 95 | 50% | 50% |
| DNp13 | 887195902 | 3 | 84 | 53 | 33% | 33% |

do not expect this to hold for every neuron. It remains likely that expanding the MCFO collection with more samples or more drivers would improve the chances of obtaining a good set of matches.

We extended the PPPM reverse analysis in *Mais et al., 2021* with a comparison to CDM (*Table 2*). We examined nine Hemibrain bodies, each with 2–13 published split-GAL4 associations (*Schretter et al., 2020*; *Wang et al., 2020b*). The best rank of each known-matching line was recorded, with *Table 2* showing the median line rank and the percentage of lines with ranks in the top 50 results. PPPM and CDM both had median line ranks under 100 for most EM bodies. PPPM was somewhat more consistent, with 33–80% of known matches in the top 50 results, compared to 0–100% for CDM. As with the forward analysis, each approach performed better on some neurons than the other approach.

## Discussion

We have described an extensive MCFO image resource from Generation 1 GAL4 lines, providing single-cell-level resolution of the neurons labeled by each line. The NeuronBridge website allows rapid searching of this resource from published EM datasets or uploaded images. CDM and PPPM search approaches both find valid EM/LM matches for several tested neurons, supporting their effectiveness and the good coverage of the brain by the Gen1 MCFO collection. NeuronBridge has already seen frequent usage (*Bidaye et al., 2020*; *Morimoto et al., 2020*; *Nojima et al., 2021*; *Sareen et al., 2021*; *Zolin et al., 2021*; *Israel et al., 2022*; *Tanaka and Clark, 2022*; *Laturney et al., 2022*). Together these tools allow for the rapid determination of likely split-GAL4 lines and other enhancer-based approaches to target most neurons initially found in the FlyEM Hemibrain and eventually in the full *Drosophila* CNS.

While performing these analyses and practically applying the tools to screen split-GAL4 combinations, we made some qualitative observations: (1) In general, both CDM and PPPM are complimentary and best used in combination, although PPPM tended to bring good matches closer to the top of search results. (2) CDM occasionally struggled with occluded neurons and benefited from examination of full 3D stacks of matching MCFO samples. (3) PPPM correspondingly showed the most improvement in samples with occluded neurons. (4) Both techniques return some highly ranked false positives with clear flaws, such that rankings alone are insufficient for algorithmic association of EM and LM neurons. (5) We estimate the image collection and search techniques can lead to good split

combinations for 50–80% of cell types, depending on how clean a combination is needed. More split hemidrivers would likely be needed to increase this rate. The search techniques do not significantly change which cell types can be targeted, but greatly simplify identifying candidate split combinations without requiring as much anatomical expertise.

There are several caveats for why close EM/LM matches do not always lead to successful split-GAL4 combinations: (1) Many CNS cell types contain multiple neurons that are indistinguishable based on morphology. Thus, two matches for a cell type may label different neurons within the cell type and fail as a split combination. Information from connectomic approaches and other modalities are also continuing to refine cell type definitions. (2) Although split-GAL4 hemidrivers are made with the same enhancer fragments as Gen1 GAL4 lines, they can differ in vector sequence and genomic insertion site. These differences can alter expression patterns and hence split-GAL4 effectiveness. (3) UAS reporters can vary in genomic insertion site, number of UAS elements, and other factors that affect how well they label particular cell types. MCFO reporters in particular can tend to brightly label neurons that are weakly labeled by reporters for the full GAL4 pattern. An examination of the full Gen1 GAL4 patterns (if not too dense) can help predict likely effectiveness of a split combination. (4) GAL4 driver expression can vary temporally, so there could be spatial but not temporal overlap between two split hemidrivers.

In creating the image resource, we have optimized driver line selection, sample preparation, and imaging to yield the maximum identifiable neurons per sample, per line, and across the central brain and VNC. For the search resource, we have implemented two complementary search approaches that effectively identify neuron matches in an easy to use interface. The image resource should be amenable to analysis with future search approaches as they continue to develop.

While our focus has been on the EM to split-GAL4 use case, we described other uses, including guiding EM proofreading and extending EM analyses beyond limited regions or sample sizes currently available. We anticipate other uses will be found for this resource.

# Materials and methods
## Fly stocks

The 5155 Generation 1 GAL4 stocks included in this resource (*Supplementary file 1*) were from *Jenett et al., 2012*; *Tirian and Dickson, 2017*. Lines in the 20×/63× (Annotator) collection were selected by collaborators for individual projects. For the 40× collection we focused on driver lines with available AD or DBD hemidrivers (*Tirian and Dickson, 2017*; *Dionne et al., 2018*). Split-GAL4 stock MB310C consists of *R52G04-p65ADZp in VK00027* and *R17C11-ZpGdbd in attP2* (*Aso et al., 2014b*). UAS reporters are described in key resources table. 'R57C10-Flp MCFO' in the text was JRC stock 3023701 for 94% of such samples, and JRC stock 3023700 for 6% of samples from sparser lines. 'hs-Flp MCFO' was JRC stock 3023951. See *Supplementary file 1* for details of individual samples.

## Key resources table

| Reagent type (species) or resource | Designation | Source or reference | Identifiers | Additional information |
|---|---|---|---|---|
| Genetic reagent (*Drosophila melanogaster*) | MCFO-1; hsPESTOPT_attP3_ 3stop1_X_0036; (w, pBPhsFlp2::PEST in attP3;; pJFRC201-10XUASFRT >STOP > FRT-myr::smGFP-HA in VK00005,pJFRC240- 10XUAS-FRT>STOP > FRT-myr::smGFP-V5-THS-10XUASFRT >STOP > FRT-myr::smGFPFLAG in su(Hw)attP1/TM3,Sb) | *Nern et al., 2015* | RRID:BDSC_64085 (Janelia stock 1117734) | |
| Genetic reagent (*Drosophila melanogaster*) | MCFO-2; pBPhsFLP_PEST_ HAV5_FLAG_OLLAS_ X3_0095; (w, pBPhsFlp2::PEST in attP3;; pJFRC210-10XUASFRT >STOP > FRT-myr::smGFP-OLLAS in attP2, pJFRC201- 10XUAS-FRT>STOP > FRT-myr::smGFP-HA in VK0005, pJFRC240-10XUAS-FRT>STOP > FRT-myr::smGFP-V5-THS10XUAS-FRT>STOP > FRT-myr::smGFPFLAG in su(Hw)attP1/TM2) | *Nern et al., 2015* | RRID:BDSC_64086 (Janelia stock 3022015) | |

*Continued on next page*

*Continued*

| Reagent type (species) or resource | Designation | Source or reference | Identifiers | Additional information |
|---|---|---|---|---|
| Genetic reagent (*Drosophila melanogaster*) | MCFO-4; 57C10wt_attp8_ 3stop1; (w, R57C10-Flp2 in su(Hw)attP8;; pJFRC201-10XUASFRT >STOP > FRT-myr::smGFP-HA in VK00005,pJFRC240- 10XUAS-FRT>STOP > FRT-myr::smGFP-V5-THS-10XUASFRT >STOP > FRT-myr::smGFP-FLAG in su(Hw)attP1) | *Nern et al., 2015* | RRID:BDSC_64088 (Janelia stock 1116898) | |
| Genetic reagent (*Drosophila melanogaster*) | MCFO-5; 57C10PEST_attp8_ 3stop1; (w, R57C10-Flp2::PEST in su(Hw)attP8;; pJFRC201- 10XUAS-FRT>STOP > FRT-myr::smGFPHA in VK00005, pJFRC240-10XUAS-FRT>STOP > FRTmyr::smGFP-V5-THS-10XUAS-FRT>STOP > FRTmyr::smGFP-FLAG in su(Hw)attP1/TM2) | *Nern et al., 2015* | RRID:BDSC_64089 (Janelia stock 1116876) | |
| Genetic reagent (*Drosophila melanogaster*) | MCFO-6; 57C10L_attp8_ 4stop1; (w, R57C10-FlpL in su(Hw)attp8;; pJFRC210-10XUASFRT >STOP > FRT-myr::smGFP-OLLAS in attP2, pJFRC201- 10XUAS-FRT>STOP > FRT-myr::smGFP-HA in VK00005, pJFRC240-10XUAS-FRT>STOP > FRT-myr::smGFP-V5-THS10XUAS-FRT>STOP > FRT-myr::smGFPFLAG in su(Hw)attP1/TM2) | *Nern et al., 2015* | RRID:BDSC_64090 (Janelia stock 1116894) | |
| Genetic reagent (*Drosophila melanogaster*) | MCFO-7; 57C10PEST_attp18_ 4stop1; (w, R57C10-Flp2::PEST in attp18;; pJFRC210-10XUASFRT >STOP > FRT-myr::smGFP-OLLAS in attP2, pJFRC201-10XUAS-FRT>STOP > FRT-myr::smGFPHA in VK00005, pJFRC240-10XUAS-FRT>STOP > FRTmyr::smGFP-V5-THS-10XUAS-FRT>STOP > FRTmyr::smGFP-FLAG in su(Hw)attP1/TM2) | *Nern et al., 2015* | RRID:BDSC_64091 (Janelia stock 1116875) | |
| Genetic reagent (*Drosophila melanogaster*) | MCFO-3 derivative; 57C10L_brp_SNAP_ MCFO_X23_0117; (w, R57C10-FlpL in su(Hw)attP8; brp::Snap / CyO; pJFRC201-10XUAS-FRT>STOP > FRT-myr::smGFPHA in VK00005,pJFRC240-10XUAS-FRT>STOP > FRTmyr::smGFP-V5-THS-10XUAS-FRT>STOP > FRTmyr::smGFP-FLAG in su(Hw)attP1/TM6B) | *Nern et al., 2015*; *Kohl et al., 2014* | RRID:BDSC_64087 (Janelia stock 3023700) | |
| Genetic reagent (*Drosophila melanogaster*) | 57C10PEST_brp_SNAP_ MCFO_X23_0099; (w, R57C10- Flp2::PEST in attP18; brp::Snap / CyO; pJFRC201-10XUASFRT >STOP > FRT-myr::smGFP-HA in VK00005,pJFRC240- 10XUAS-FRT>STOP > FRT-myr::smGFP-V5-THS-10XUASFRT >STOP > FRT-myr::smGFP-FLAG in su(Hw)attP1/TM6B) | *Nern et al., 2015* | (Janelia stock 3023701) | |
| Genetic reagent (*Drosophila melanogaster*) | MCFO-1 derivative; pBPhsFlp2_PEST_ brp_SNAP_ MCFO_0128; (w, pBPhsFlp2::PEST in attP3; brp::Snap / CyO; pJFRC201- 10XUAS-FRT>STOP > FRT-myr::smGFPHA in VK00005,pJFRC240-10XUAS-FRT>STOP > FRTmyr::smGFP-V5-THS-10XUAS-FRT>STOP > FRTmyr::smGFP-FLAG in su(Hw)attP1/TM6B) | *Nern et al., 2015*; *Kohl et al., 2014* | RRID:BDSC_64085 (Janelia stock 3023951) | |
| Genetic reagent (*Drosophila melanogaster*) | pJFRC2-10XUAS-IVS-mCD8::GFP | *Pfeiffer et al., 2010* | RRID:BDSC_32185 (Janelia stock 1115125) | |
| Antibody | Anti-Brp mouse monoclonal nc82 | Developmental Studies Hybridoma Bank (DSHB) | RRID: AB_2314866 | 1:30 |
| Antibody | Anti-HA rabbit monoclonal C29F4 | Cell Signaling Technologies: 3724S | RRID: AB_1549585 | 1:300 |
| Antibody | Anti-FLAG rat monoclonal DYKDDDDK Epitope Tag Antibody | Novus Biologicals: NBP1-06712 | RRID: AB_1625981 | 1:200 |
| Antibody | DyLight 550 conjugated anti-V5 mouse monoclonal | AbD Serotec: MCA1360D550GA | RRID: AB_2687576 | 1:500 |
| Antibody | Anti-RAT IgG (H&L) Goat Polyclonal Antibody ATTO 647N Conjugated | Rockland: 612-156-120 | RRID: AB_10893386 | 1:300 |
| Antibody | Alexa Fluor 594 AffiniPure Donkey Polyclonal Anti-Rabbit IgG (H+L) | Jackson ImmunoResearch Labs: 711-585-152 | RRID: AB_2340621 | 1:500 |

*Continued on next page*

*Continued*

| Reagent type (species) or resource | Designation | Source or reference | Identifiers | Additional information |
|---|---|---|---|---|
| Antibody | Anti-Green Fluorescent Protein (GFP) Rabbit Polyclonal Antibody, Unconjugated | Thermo Fisher Scientific: A-11122 | RRID: AB_221569 | 1:1000 |
| Antibody | Goat Polyclonal anti-Rabbit IgG (H+L) Highly Cross-Adsorbed Antibody, Alexa Fluor 488 | Thermo Fisher Scientific: A-11034 | RRID: AB_2576217 | 1:800 |
| Antibody | Goat Polyclonal anti-Mouse IgG (H+L) Highly Cross-Adsorbed Antibody, Alexa Fluor 568 | Thermo Fisher Scientific: A-11031 | RRID: AB_144696 | 1:800 |
| Software, algorithm | Janelia Workstation | *Rokicki et al., 2019*; https://github.com/JaneliaSciComp/workstation; *Howard Hughes Medical Institute, 2023* | RRID: SCR_014302 | |
| Software, algorithm | NeuronBridge codebase | *Clements et al., 2022*; *Clements et al., 2021* https://doi.org/10.25378/janelia.12159378.v2 | | |
| Software, algorithm | Fiji | https://fiji.sc | RRID: SCR_0022852 | |
| Software, algorithm | Affinity Designer | https://affinity.serif.com/designer/ | RRID: SCR_016952 | |
| Other | MCFO Hybrid Chemical Tag & IHC for Adult CNS | https://doi.org/10.17504/protocols.io.nyhdft6 | | Protocol |
| Other | FlyLight protocols for dissection, immunohistochemistry, and mounting | https://www.janelia.org/project-team/flylight/protocols | | Protocol |

## Fly crosses, heat shock, and dissection

Flies were raised on standard corn meal molasses food, typically in at least partial-brightness 24 hr light. All crosses were performed at 21–25°C, with a few exceptions (~2.5% of all samples) performed at 18°C when scheduling necessitated. Crosses with hs-Flp in particular were held at 21°C until adulthood, when they were heat shocked at 37°C for 40 min (Category 2 lines) or 13 min (Category 3 lines). Flies were generally dissected at 5–14 days of adulthood, giving time for R57C10-Flp and then MCFO reporter expression.

## Tagging and immunohistochemistry

After dissection of the brain or full CNS, samples were fixed for 55 min in 2% paraformaldehyde.

For the 40× pipeline a hybrid labeling protocol was used, in which a chemical tag (Brp-SNAP and SNAP-tag ligand) labels the neuropil reference, and immunohistochemistry of MCFO markers labels specific GAL4 neurons (*Kohl et al., 2014*; *Nern et al., 2015*; *Meissner et al., 2018*). See Key resources table for specific antibodies and concentrations. Chemical tag labeling of the Brp reference was not as bright as Brp antibody staining with nc82, but was more consistent and had lower background. 40× pipeline samples were washed one to four times for 15 min and then tagged with 2 µM

Cy2 SNAP-tag ligand to visualize the Brp-SNAP neuropil the same day, after which immunohistochemistry and DPX mounting followed.

20×/63× samples used nc82 for neuropil reference labeling, as in *Nern et al., 2015*, and typically received four washes of 10 min each after fixation. See https://www.janelia.org/project-team/flylight/protocols for full MCFO protocols with either nc82 or hybrid Brp-SNAP neuropil labeling.

## Imaging and image processing

Imaging was performed using eight Zeiss LSM 710 or 780 laser scanning confocal microscopes over a combined capture time of 11 years. 20×/63× imaging was performed with 20× air and 63× oil objectives to combine rapid scanning of all samples with detailed scanning of regions of interest. 40× imaging was performed with 40× oil objectives to cover the central brain and VNC with good axial resolution in a single pass. Confocal stacks were captured at 0.52 × 0.52 × 1.00 µm (20× objective), 0.19 × 0.19 × 0.38 µm (63×), or 0.44 µm isotropic resolution (40×). 40× resolution was selected to maximize effective *z*-resolution while limiting the size of the full dataset (about 100 TB combined). The field of view was set to the widest 0.7 zoom for 40× and 63× objectives, resulting in heightened lens distortion at the edges of images, which was corrected before stitching (*Bogovic et al., 2020*). The whole brain and VNC (where present) were captured in separate 20× tiles for 20×/63× samples, followed by selected 63× tiles of regions of interest. The central brain and two VNC tiles (where present) were captured for each 40× sample. After merging and distortion correction, overlapping 40×/63× tiles were automatically stitched together, as described (*Yu and Peng, 2011*). Brains and VNCs were aligned to the JRC2018 sex-specific and unisex templates using CMTK software, and color depth MIPs were generated (*Rohlfing and Maurer, 2003*; *Otsuna et al., 2018*; *Bogovic et al., 2020*).

Four-color imaging was configured as described in *Nern et al., 2015*. Briefly, two LSM confocal stacks were captured at each location, one with 488 and 594 nm laser lines and one with 488, 561, and 633 nm laser lines. Stacks were merged together after imaging. Imaging was performed using Zeiss's ZEN software with a custom MultiTime macro. The macro was programmed to automatically select appropriate laser power for each sample and region, resulting in independent image parameters between samples and between brains and VNCs. Gain was typically set automatically for the 561 and 633 nm channels and manually for 488 and 594 nm. Imaging parameters were held constant within tiles covering a single brain or VNC.

The image processing pipeline (distortion correction, normalization, merging, stitching, alignment, MIP generation, file compression) was automated using the open-source Janelia Workstation software (*Rokicki et al., 2019*), which was also used to review the secondary results and annotate lines for publishing. Images for published lines were uploaded to AWS S3 (Amazon Web Services) and made available in a public bucket (https://registry.opendata.aws/janelia-flylight/) for download or further analysis on AWS. Original LSM (i.e., lossless TIFF) imagery is available alongside the processed (merged/stitched/aligned) imagery in H5J format. H5J is a 'visually lossless' format developed at Janelia, which uses the H.265 codec and differential compression ratios on a per-channel basis to obtain maximum compression while minimizing visually relevant artifacts (see http://data.janelia.org/h5j).

The open-source NeuronBridge tool (*Clements et al., 2021*; *Clements et al., 2022*) is a web application designed for ease of use and accessibility to neuron mappings across large multi-modal datasets. It hosts precomputed matches for publicly available EM and LM datasets originating at Janelia, and also supports ad hoc searches against those datasets based on user data. NeuronBridge was constructed as a single-page application built on the React framework for fast performance, responsiveness, and ease of deployment. The web app and backend services are both deployed to AWS to ensure scalability and reliability, and they use only serverless components to minimize costs. NeuronBridge also takes advantage of the innovative 'burst-parallel' compute paradigm (*Fouladi et al., 2019*) to massively scale color depth MIP search by leveraging micro VMs (virtual machines) on AWS Lambda, thereby enabling rapid ad hoc searches across a nominally petabyte-scale dataset.

## Quality control and expression density categorization

Samples had to pass quality control at several stages to be included in the final collection. Samples lacking visible neuron expression or too dense for IHC were in most cases excluded prior to imaging. Samples were excluded that contained damage, distortion, debris, or low neuropil reference quality

causing a failure to align or an error in the image processing pipeline. Samples with minor issues in neuron channels were typically included if neurons could be distinguished. Every effort was made to accurately track and correct line and sample metadata, but the dataset may still contain occasional errors.

Selected *Drosophila* lines were qualitatively grouped into Categories 1 through 5 by expression density, primarily using MCFO and less often by full GFP patterns. Category boundaries were selected based on our estimation of the utility of the lines and their anticipated performance for neuron segmentation. Category 1 and 5 samples were excluded due to lack of information, either no unique neurons or too many to label, respectively. Categories 3 and 4 were divided based on estimated difficulty of manual segmentation combined with intuition about future segmentation algorithm improvements, such that Category 3 lines are expected to be tractable for segmentation, whereas Category 4 lines are more challenging. Categories 2 and 3 were divided such that Category 2 mostly contained neurons that could easily be 'segmented' by eye, whereas Category 3 had more instances of overlapping neurons that were harder to distinguish.

### Search approach evaluation

For the forward analysis, the top 100 NeuronBridge search results were examined for one Hemibrain body in each cell type. About 20% of the samples were checked by opening the image stacks, including the majority of the samples annotated as including the cell type in question.

Reverse analysis was performed as in *Mais et al., 2021*.

### Acknowledgements

This work is part of the FlyLight Project Team at Janelia Research Campus, Howard Hughes Medical Institute, Ashburn, VA. Author order includes the following alphabetical groups: FlyLight Project Team, Janelia Scientific Computing Shared Resource, and contributing laboratories. During this effort, the FlyLight Project Team included Megan Atkins, Shelby Bowers, Kari Close, Gina DePasquale, Zack Dorman, Kaitlyn Forster, Jaye Anne Gallagher, Theresa Gibney, Asish Gulati, Joanna Hausenfluck, Yisheng He, Kristin Hendersen, Hsing Hsi Li, Nirmala Iyer, Jennifer Jeter, Lauren Johnson, Rebecca Johnston, Rachel Lazarus, Kelley Lee, Hua-Peng Liaw, Oz Malkesman, Geoffrey Meissner, Brian Melton, Scott Miller, Reeham Motaher, Alexandra Novak, Omatara Ogundeyi, Alyson Petruncio, Jacquelyn Price, Sophia Protopapas, Susana Tae, Athreya Tata, Jennifer Taylor, Allison Vannan, Rebecca Vorimo, Brianna Yarborough, Kevin Xiankun Zeng, and Chris Zugates, with Steering Committee of Yoshinori Aso, Gwyneth Card, Barry Dickson, Reed George, Wyatt Korff, Gerald Rubin, and James Truman. We thank Gudrun Ihrke and Project Technical Resources for management coordination and staff support. We thank Melanie Radcliff for administrative support. We thank Barret Pfeiffer for his early work in developing the MCFO method. We thank Teri Ngo for her early collaborations with FlyLight. We thank Kei Ito, Kristin Scott, and Michael H Dickinson for contributions to visitor and team projects. For setting up thousands of crosses, we thank the Janelia Fly Facility: Amanda Cavallaro, Tam Dang, Guillermo Gonzalez, Scarlett Harrison, Jui-Chun Kao, Todd R Laverty, Brenda Perez, Brandi Sharp, Viruthika Vallanadu, and Grace Zheng. We thank Karen Hibbard for establishing the brp-SNAP MCFO reporter stocks. We thank Mark Bolstad, Tom Dolafi, Leslie L Foster, Sean Murphy, Donald J Olbris, Todd Safford, Eric Trautman, and Yang Yu for their work on software infrastructure. We thank Ruchi Parekh and Stephen M Plaza for EM/LM coordination. Stocks obtained from the Bloomington *Drosophila* Stock Center (NIH P40OD018537) were used in this study. We thank them, especially Annette Parks, Cale Whitworth, and Sam Zheng, for the maintenance and distribution of stocks from the Janelia collection. Funding was provided by Howard Hughes Medical Institute. This article is subject to HHMI's Open Access to Publications policy. HHMI lab heads and project team leads have previously granted a nonexclusive CC BY 4.0 license to the public and a sublicensable license to HHMI in their research articles. Pursuant to those licenses, the author-accepted manuscript of this article can be made freely available under a CC BY 4.0 license immediately upon publication.

# Additional information

## Competing interests

FlyLight Project Team: The other authors declare that no competing interests exist.

## Funding

| Funder | Grant reference number | Author |
|---|---|---|
| Howard Hughes Medical Institute | | Geoffrey W Meissner<br>Aljoscha Nern<br>Yoshinori Aso<br>Gwyneth M Card<br>Barry J Dickson<br>Wyatt Korff<br>Gerald M Rubin<br>FlyLight Project Team |

The funders had no role in study design, data collection, and interpretation, or the decision to submit the work for publication.

## Author contributions

Geoffrey W Meissner, Conceptualization, Data curation, Formal analysis, Supervision, Funding acquisition, Validation, Investigation, Visualization, Methodology, Writing – original draft, Project administration, Writing – review and editing; Aljoscha Nern, Conceptualization, Data curation, Formal analysis, Validation, Investigation, Visualization, Methodology, Writing – original draft, Writing – review and editing; Zachary Dorman, Data curation, Formal analysis, Validation, Investigation, Visualization, Methodology, Writing – original draft, Project administration, Writing – review and editing; Gina M DePasquale, Kaitlyn Forster, Theresa Gibney, Yisheng He, Lauren Johnson, Brianna Yarbrough, Validation, Investigation; Joanna H Hausenfluck, Erica Ehrhardt, Jens Goldammer, Masayoshi Ito, Ryo Minegishi, Shigehiro Namiki, Investigation; Nirmala A Iyer, Validation, Investigation, Methodology; Jennifer Jeter, Data curation, Validation, Investigation, Visualization, Methodology; Rebecca M Johnston, Data curation, Supervision, Validation, Investigation, Methodology, Project administration; Kelley Lee, Data curation, Validation, Investigation, Project administration; Brian Melton, Data curation, Validation, Investigation, Methodology; Christopher T Zugates, Supervision, Funding acquisition, Project administration; Jody Clements, Cristian Goina, Software, Visualization; Hideo Otsuna, Software, Formal analysis, Visualization, Methodology; Konrad Rokicki, Data curation, Software, Supervision, Funding acquisition, Validation, Visualization, Methodology, Writing – original draft, Project administration, Writing – review and editing; Robert R Svirskas, Data curation, Software, Validation; Yoshinori Aso, Conceptualization, Investigation, Methodology, Writing – review and editing; Gwyneth M Card, Conceptualization, Supervision; Barry J Dickson, Conceptualization, Supervision, Writing – review and editing; Dagmar Kainmueller, Conceptualization, Software, Formal analysis, Supervision, Validation, Visualization, Methodology, Writing – review and editing; Wyatt Korff, Oz Malkesman, Conceptualization, Supervision, Funding acquisition, Methodology, Project administration; Lisa Mais, Software, Formal analysis, Validation, Visualization, Methodology; Gerald M Rubin, Conceptualization, Supervision, Funding acquisition, Methodology, Writing – review and editing; Gabriella R Sterne, Tanya Wolff, Investigation, Writing – review and editing; FlyLight Project Team, Conceptualization, Data curation, Funding acquisition, Investigation, Methodology, Project administration, Supervision, Validation, Visualization, Writing – original draft, Writing – review and editing

## Author ORCIDs

Geoffrey W Meissner ⓘ http://orcid.org/0000-0003-0369-9788
Aljoscha Nern ⓘ http://orcid.org/0000-0002-3822-489X
Zachary Dorman ⓘ http://orcid.org/0000-0001-9933-7217
Theresa Gibney ⓘ http://orcid.org/0000-0001-5461-724X
Christopher T Zugates ⓘ http://orcid.org/0000-0003-1882-3665
Cristian Goina ⓘ http://orcid.org/0000-0003-2835-7602
Hideo Otsuna ⓘ http://orcid.org/0000-0002-2107-8881
Konrad Rokicki ⓘ http://orcid.org/0000-0002-2799-9833
Robert R Svirskas ⓘ http://orcid.org/0000-0001-8374-6008

Yoshinori Aso http://orcid.org/0000-0002-2939-1688
Gwyneth M Card http://orcid.org/0000-0002-7679-3639
Barry J Dickson http://orcid.org/0000-0003-0715-892X
Erica Ehrhardt http://orcid.org/0000-0002-9252-1414
Jens Goldammer http://orcid.org/0000-0002-5623-8339
Dagmar Kainmueller http://orcid.org/0000-0002-9830-2415
Wyatt Korff http://orcid.org/0000-0001-8396-1533
Ryo Minegishi http://orcid.org/0000-0002-2895-9438
Shigehiro Namiki http://orcid.org/0000-0003-1559-799X
Gerald M Rubin http://orcid.org/0000-0001-8762-8703
Gabriella R Sterne http://orcid.org/0000-0002-7221-648X
Tanya Wolff http://orcid.org/0000-0002-8681-1749
Oz Malkesman http://orcid.org/0000-0003-2219-7476

**Decision letter and Author response**
Decision letter https://doi.org/10.7554/eLife.80660.sa1
Author response https://doi.org/10.7554/eLife.80660.sa2

---

## Additional files

### Supplementary files

• Supplementary file 1. Generation 1 MultiColor FlpOut (MCFO) samples included in the study. Metadata for the included 74,363 MCFO samples from 5155 Gen1 GAL4 lines are tabulated, including line name, landing site, effector, slide code, creation date, GUID, gender, heat shock duration, objectives, release name, and contributing annotator. See Key resources table for effector codes.

• MDAR checklist

### Data availability

The footprint of this image resource (~105 TB) exceeds our known current practical limits on standard public data repositories. Thus, we have made all the primary data (and a variety of processed outputs) used in this study freely available under a CC BY 4.0 license at https://doi.org/10.25378/janelia.21266625.v1 and through the publicly accessible website https://gen1mcfo.janelia.org. The images are made searchable with the same permissions on the user-friendly NeuronBridge website https://neuronbridge.janelia.org. NeuronBridge code is available at *Clements et al., 2021* and the application and implementation are discussed further in *Clements et al., 2022*.All other data generated or analyzed during this study are included in the manuscript and supporting files.

The following dataset was generated:

| Author(s) | Year | Dataset title | Dataset URL | Database and Identifier |
|---|---|---|---|---|
| Svirskas R, Rokicki K, Bates A, Svirskas R | 2023 | Fly Brain Anatomy: FlyLight Gen1 and Split-GAL4 Imagery | https://doi.org/10.25378/janelia.21266625.v1 | Registry of Open Data on AWS, 10.25378/janelia.21266625.v1 |

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
