## [Editor Report]

This study bridges the gap between connectomic data from the fly hemibrain and driver lines needed for functional experiments. The large collection of labeled single cells clones from a large number of samples now provides the community to search both light microscopic and electron microscopic databases for matches using single cells, or cell types. Overall, this manuscript does a compelling job of describing an important resource for the community, which will hopefully be built upon via the collaborative science of many groups as the field develops.

---

## [Decision Letter]

**Decision letter after peer review:**

Thank you for submitting your article "A searchable image resource of *Drosophila* GAL4-driver expression patterns with single neuron resolution" for consideration by *eLife*. Your article has been reviewed by 2 peer reviewers, and the evaluation has been overseen by a Reviewing Editor and Claude Desplan as the Senior Editor. The reviewers have opted to remain anonymous.

Essential revisions:

As you will see from the detailed reviews, the reviewers found the tools and resource described in your manuscript to be highly valuable to the fly community. Nevertheless, they request and recommend a number of changes to the manuscript and figures (and possibly database) to make the tool more widely accessible and user friendly. Please provide detailed answers to all points below and address in particular the points mentioned in 'recommendations to the authors' by providing additional explanation and/or editing the manuscript text.

As an additional note, one of the reviewers mentions the discrepancy between the number of central brain neurons cited in this paper (30,000) as compared to more rigorous evaluations from the Potter and Gronenberg labs that cite much higher numbers of neurons. A recent bioRxiv paper from Seung and Murthy (bioRxiv 2021.11.04.467197; doi: https://doi.org/10.1101/2021.11.04.467197) gets closer to the number cited here. It would be important, especially in this type of paper, to be precise in citing these numbers as further papers might reference the current paper as the truth, or at least to mention the discrepancies!

*Reviewer #1 (Recommendations for the authors):*

Points that might benefit the clarity of the revised manuscript:

1. Please consider revising Figure 1 to make it more clear which images belong to which label. For example, in Figure 1A and B1, a more accurate description of the images would be helpful (it was unclear that the VNC was not a separate panel from the CNS). Moving 'Gal4' and 'MCFO' text to the bottom of the VNC might help. Also, perhaps revising Figure 1 so it was organized like more of a flow chart (a bit like Figure 3) may be helpful.

2. Please clarify what makes a brain 'aligned' for upload as a 2D MIP.

3. Please clarify what score similarity numbers mean.

4. Quotations around hybrid are not necessary (line 384).

5. Perhaps abbreviate Color Depth Maximum intensity projection (CDM) without the MIP which adds confusion (line 81).

6. Material and Methods- For the IHC section it would be helpful if Table 1 was referenced so readers would know where to find the information for the antibodies and dilutions used.

7. We appreciated methodology differences between Flp methods in the intro. One additional sentence of background introducing this method (with citations) would be helpful for non-expert readers (~lines 61-64).

8. It would be helpful if the authors elaborate a bit more on the importance of evaluating the search performance of CDM & PPPM approaches in the intro. A sentence clarifying why these are helpful would be appropriate.

9. Typo 'respectively' (line 300).

10. Perhaps add a comma at end of line 49.

11. doi for Rokicki et al. (2019) in the references is not working.

*Reviewer #2 (Recommendations for the authors):*

1) The manuscript mentions that the NeuronBridge tool will be further described in Rokicki et al., 2022 (in preparation). At the same time, the abstract says "we have also developed the NeuronBridge search tool" (lines 26-27) and the introduction "We have released the data, along with the NeuronBridge tool" (line 99). Is the Rokicki et al., 2022 the same publication as the recent preprint Clements et al., 2022 (https://doi.org/10.1101/2022.07.20.500311)?

It would be helpful if the manuscript was clear on which aspects of NeuronBridge are being described for the first time in this work versus the recent preprint. For example, if it is the integration of the MCFO dataset and/or the application of the search tools, while the development of the tool pertains to the other publication.

2) lines 182-3: the estimate for number of neurons in the central brain and VNC seems low (30k and 15k respectively). There are more recent estimates from a whole brain connectomics dataset for the brain (Mu et al., 2021, bioRxiv, https://doi.org/10.1101/2021.11.04.467197) which put this number at 43k ± 4k. The Hemibrain dataset which does not include most of the subesophageal zone, already includes 26k (truncated and non-truncated). For the VNC a personal communication in Bates et al., 2019 (https://doi.org/10.1016/j.conb.2018.12.012) suggests ~20k.

Could the mention of neuron number refer to more recent publications, or present a range?

3) line 240-1: it is not made clear from the phrasing ("currently allows") if it is expected that the NeuronBridge tool would integrate EM datasets beyond those generated by the FlyEM team. It would be helpful if this was clarified.

4) lines 270-1: regarding the option for users to upload an unregistered stack, could it be made clear if, after registration, how users can assess its quality, and thus take that into account when assessing search hits.

5) from line 286: regarding the reverse search examples, and more generally, providing the user with ways to gather more information on EM neurons or LM GAL4 or split-GAL4 lines. The NeuronBridge website displays a link from the search results to the Virtual Fly Brain website, though this mention is lacking in the manuscript. It would seem useful to add it.

6) From line 210: the sparse T neuron observed in many lines is an interesting observation. Without a comprehensive analysis to see if other neurons are commonly labeled, it is not clear if the neuron stands out simply because of its morphology. It would be helpful if it was made clear that this neuron might not be unique regarding its high frequency.

*Reviewer #3 (Recommendations for the authors):*

Detailed Comments to the Authors:

Comments on Data Pre-Processing and Annotation:

P4, l145: „GAL4 lines were qualitatively categorized by density …." How was categorization performed? Please provide mathematical details on the quantitative quality measures and the algorithm if done automatically or explain the manual process (e.g. by whom this was done). Directly related to this is a comment on the Materials and methods section (making the above comment partially redundant. I leave it to the authors where best to address this): P15, l439++:

The description of quality assurance and data categorization processes would benefit from more precision:

– Who performed the quality control and categorization? What is the professional background of these persons?

– Where any automatic tools involved in quality assurance? If yes, provide details on tools and implemented methods, and of their use.

– The provided description of the different categories is imprecise and leaves plenty room for subjective decisions.

– Were persons doing the categorization (if done manually) provided with a clear protocol on how to decide on the categories? It is e.g. not explained what kind of visualization (2D section, 2D MIP, 3D, interactive 3D?) was used to do the categorization, which might influence the decision. Did people inspect all available imaging data related to a line or did they select a specific image? If I understood this correctly, MCFO is not labeling all neurons of a line. Is it therefore sufficient to inspect only the image of a single sample to define the category of a line or are more required? If not, please explain how many were inspected per line in average?

– "Category boundaries were initially based on functional properties" What are "functional properties" in this context? Please clarify.

– You state that category 3 and category 4 were separated using the result on a neuron segmentation algorithm and intuition on future segmentation difficulty. Which segmentation algorithm? What exactly is "future segmentation difficulty" referring to?

– Category 2 – what exactly do you mean with "segmented" by eye? Based on what kind of visualization? (see above)

P7, l237: From the text, it was not clear to me if the alignment to JRC2018 included also the EM dataset. As this is essential for the whole matching process, it would be helpful to explicitly mention that alignment has been done for both, LM and EM data.

Comments on Described Search Approaches

P7, l250: The reference (Otsuna et al. 2022) seems to contain important details on preprocessing and one of the used matching algorithms. However, the paper is still unpublished and is marked in the reference list as "in preparation". For transparency and replicability reasons, the authors should ensure that algorithmic details are accessible to the public and either provide a prepublication highlighting the respective method or include method details in this paper.

P7, l244 + l255 and P15, l419: PPPM requires segmentation of neurons in LM images. However, information on how this segmentation is performed is missing. Please clarify.

P8, l269+

Providing a custom processing and search capability for private 3D image stacks is certainly a great service. However, it might also raise concerns of potential users in terms of trust in respect to potential disclosure of private research data after upload. To be transparent in this respect, it would be beneficial to add information on data handling procedures in terms of privacy and security. Simply said, as a user I would like to know what happens with my data and its derivations after upload and processing? Is it deleted? Who has access to it?

Comments on Search Approach Evaluation:

The described evaluation approach is only partially feasible to support the very general claim of the authors that "NeuronBridge rapidly and effectively identifies neuron matches …." (P1, l28; and other parts of the manuscript): Although the evaluation appears to be straight forward and comprehensive at first glance, it lacks scientific rigor in several dimensions (see also comments below) and does not follow established evaluation standards in information retrieval. I understand that a full evaluation of the system according to the state of the art might be out of scope of this paper. However, in this case the claim should highlight that the performed experiments provide a demonstration of the potential of the method to effectively identify neuron matches, but not a proof.

If the authors want to keep the generality in the claim, they should consider to revise this section in respect to the design of their experiments and the formal selection, justification and use of appropriate quality measures introduced by the information retrieval community. (The following manuscript might serve as an entry point and design further experiments: https://nlp.stanford.edu/IR-book/html/htmledition/evaluation-in-information-retrieval-1.html)

Besides of the more high-level comment above, I have some remarks that might further illustrate my concerns:

– The numbers of selected query items and performed queries are too low to draw general conclusions.

– Transparency of the experience design:

In both experiments sets of 10 and 9 neurons respectively were selected to perform query result analysis. It remains unclear based on what criteria exactly these neurons were selected, leaving open if there is any bias in the selection which would disqualify the results.

Forward, "qualitative", Analysis: A true qualitative evaluation would require the repetition of retrieval experiments with several experts and an investigation of the question if there is e.g. an inter-observer variability, which seems not to be the case here. In this context also information on the number and professional background of the persons who judged on the query results should be added.

---

## [Author Response]

Essential revisions:As you will see from the detailed reviews, the reviewers found the tools and resource described in your manuscript to be highly valuable to the fly community. Nevertheless, they request and recommend a number of changes to the manuscript and figures (and possibly database) to make the tool more widely accessible and user friendly. Please provide detailed answers to all points below and address in particular the points mentioned in 'recommendations to the authors' by providing additional explanation and/or editing the manuscript text.As an additional note, one of the reviewers mentions the discrepancy between the number of central brain neurons cited in this paper (30,000) as compared to more rigorous evaluations from the Potter and Gronenberg labs that cite much higher numbers of neurons. A recent bioRxiv paper from Seung and Murthy (bioRxiv 2021.11.04.467197; doi: https://doi.org/10.1101/2021.11.04.467197) gets closer to the number cited here. It would be important, especially in this type of paper, to be precise in citing these numbers as further papers might reference the current paper as the truth, or at least to mention the discrepancies!

We thank the reviewers for pointing out this issue and their comments in general. We now mention the different recently reported cell counts in the text:

Line 182. Although recent estimates vary (37k to 100k neurons for the central brain including subesophageal ganglion but not the optic lobes, 15k to 20k for the VNC (Bates 2019, Godfrey 2021, Mu 2021, Raji 2021)), the adult *Drosophila* central brain has many more neurons than the VNC, suggesting earlier diminishing returns in the VNC.

Reviewer #1 (Recommendations for the authors):Points that might benefit the clarity of the revised manuscript:1. Please consider revising Figure 1 to make it more clear which images belong to which label. For example, in Figure 1A and B1, a more accurate description of the images would be helpful (it was unclear that the VNC was not a separate panel from the CNS). Moving 'Gal4' and 'MCFO' text to the bottom of the VNC might help. Also, perhaps revising Figure 1 so it was organized like more of a flow chart (a bit like Figure 3) may be helpful.

Reorganized Figure 1. Replaced split-GAL4 image with a different sample from the same line to illustrate full CNS.

2. Please clarify what makes a brain 'aligned' for upload as a 2D MIP.

This will be addressed on the revised website UI.

3. Please clarify what score similarity numbers mean.

This is already explained by the help page at https://neuronbridge.janelia.org/help, and we are adding more details to the explanation.

4. Quotations around hybrid are not necessary (line 384).

Line 384. Removed quotation marks.

5. Perhaps abbreviate Color Depth Maximum intensity projection (CDM) without the MIP which adds confusion (line 81).

Line 81. Changed to "Color Depth Maximum intensity projection (CDM) search"

6. Material and Methods- For the IHC section it would be helpful if Table 1 was referenced so readers would know where to find the information for the antibodies and dilutions used.

Line 386. Added "See Table 1 for specific antibodies and concentrations."

7. We appreciated methodology differences between Flp methods in the intro. One additional sentence of background introducing this method (with citations) would be helpful for non-expert readers (~lines 61-64).

Line 65. Added sentences "In brief, MCFO can use several UAS reporters that are independently stochastically activated by low levels of Flp recombinase. Flp levels can be adjusted to tailor MCFO labeling density for different GAL4 lines or purposes."

8. It would be helpful if the authors elaborate a bit more on the importance of evaluating the search performance of CDM & PPPM approaches in the intro. A sentence clarifying why these are helpful would be appropriate.

Line 86. Added sentence "Overlapping neurons remain challenging to segment manually or algorithmically, making this an area of rapid development. "

9. Typo 'respectively' (line 300).

Line 300. Fixed typo in "respectively".

10. Perhaps add a comma at end of line 49.

Line 49. Added comma.

11. doi for Rokicki et al. (2019) in the references is not working.

We agree that all DOI links are not being output correctly. It's not clear whether it's a limitation of the publishing system, but we'll try to remove incorrect links.

Reviewer #2 (Recommendations for the authors):1) The manuscript mentions that the NeuronBridge tool will be further described in Rokicki et al., 2022 (in preparation). At the same time, the abstract says "we have also developed the NeuronBridge search tool" (lines 26-27) and the introduction "We have released the data, along with the NeuronBridge tool" (line 99). Is the Rokicki et al., 2022 the same publication as the recent preprint Clements et al., 2022 (https://doi.org/10.1101/2022.07.20.500311)?It would be helpful if the manuscript was clear on which aspects of NeuronBridge are being described for the first time in this work versus the recent preprint. For example, if it is the integration of the MCFO dataset and/or the application of the search tools, while the development of the tool pertains to the other publication.

The intention is for this paper to cover the usage of the NeuronBridge website to search the released data. Rokicki 2022 covers the details of the NeuronBridge application and development.

Line 26. Changed text "also developed" to "made the images searchable on".

Line 99. Replaced sentence with "We have released the image data and made it searchable on the NeuronBridge website together with data from the FlyEM hemibrain and published split-GAL4 lines."

2) lines 182-3: the estimate for number of neurons in the central brain and VNC seems low (30k and 15k respectively). There are more recent estimates from a whole brain connectomics dataset for the brain (Mu et al., 2021, bioRxiv, https://doi.org/10.1101/2021.11.04.467197) which put this number at 43k ± 4k. The Hemibrain dataset which does not include most of the subesophageal zone, already includes 26k (truncated and non-truncated). For the VNC a personal communication in Bates et al., 2019 (https://doi.org/10.1016/j.conb.2018.12.012) suggests ~20k.Could the mention of neuron number refer to more recent publications, or present a range?

Please see response under Essential Revisions.

3) line 240-1: it is not made clear from the phrasing ("currently allows") if it is expected that the NeuronBridge tool would integrate EM datasets beyond those generated by the FlyEM team. It would be helpful if this was clarified.

We will consider adding other data sets in the future, but for now are focused on adding new FlyLight/FlyEM data sets as they become available.

4) lines 270-1: regarding the option for users to upload an unregistered stack, could it be made clear if, after registration, how users can assess its quality, and thus take that into account when assessing search hits.

The revised website will (a) show an alignment score for each uploaded stack, and (b) allow the user to view the aligned reference channel overlaid on the alignment template, to allow for visual assessment of the registration.

5) from line 286: regarding the reverse search examples, and more generally, providing the user with ways to gather more information on EM neurons or LM GAL4 or split-GAL4 lines. The NeuronBridge website displays a link from the search results to the Virtual Fly Brain website, though this mention is lacking in the manuscript. It would seem useful to add it.

Line 268. Added "Search results are linked directly to corresponding data in other online resources such as Virtual Fly Brain" with citation.

6) From line 210: the sparse T neuron observed in many lines is an interesting observation. Without a comprehensive analysis to see if other neurons are commonly labeled, it is not clear if the neuron stands out simply because of its morphology. It would be helpful if it was made clear that this neuron might not be unique regarding its high frequency.

Line 218. Added "No other neurons were observed to be so frequently labeled."

Reviewer #3 (Recommendations for the authors):Detailed Comments to the Authors:Comments on Data Pre-Processing and Annotation:P4, l145: „GAL4 lines were qualitatively categorized by density …." How was categorization performed? Please provide mathematical details on the quantitative quality measures and the algorithm if done automatically or explain the manual process (e.g. by whom this was done). Directly related to this is a comment on the Materials and methods section (making the above comment partially redundant. I leave it to the authors where best to address this): P15, l439++:The description of quality assurance and data categorization processes would benefit from more precision:– Who performed the quality control and categorization? What is the professional background of these persons?– Where any automatic tools involved in quality assurance? If yes, provide details on tools and implemented methods, and of their use.– The provided description of the different categories is imprecise and leaves plenty room for subjective decisions.– Were persons doing the categorization (if done manually) provided with a clear protocol on how to decide on the categories? It is e.g. not explained what kind of visualization (2D section, 2D MIP, 3D, interactive 3D?) was used to do the categorization, which might influence the decision. Did people inspect all available imaging data related to a line or did they select a specific image? If I understood this correctly, MCFO is not labeling all neurons of a line. Is it therefore sufficient to inspect only the image of a single sample to define the category of a line or are more required? If not, please explain how many were inspected per line in average?– "Category boundaries were initially based on functional properties" What are "functional properties" in this context? Please clarify.– You state that category 3 and category 4 were separated using the result on a neuron segmentation algorithm and intuition on future segmentation difficulty. Which segmentation algorithm? What exactly is "future segmentation difficulty" referring to?– Category 2 – what exactly do you mean with "segmented" by eye? Based on what kind of visualization? (see above)

The reviewer raised a number of concerns related to quality control and qualitative analysis. Quality control was performed by most or all coauthors at many stages of data generation and review. In most cases it was a manual process integrated into routine work and data analysis, such as excluding an image that does not show any labeling or that could not be adequately registered. As described, we also examined the data set as a whole once complete.

We used the term "qualitative" to convey a manual analysis based on the judgement of the investigator, rather than in the formal context of the field of information retrieval. While we believe most readers of the paper would interpret our usage correctly, we have made several modifications below for clarity.

We don't feel that the professional background of the person performing the analysis is a relevant detail to include for the quality control and density categorization. In addition, the density categorization only served to facilitate decisions on which labeling conditions to focus on in view of limited resources but does not directly influence subsequent uses of the data (such as searches for neurons of interest).

Changes:

Line 145. Added "into rough groups".

Line 448. Changed "Category boundaries were initially established based on functional properties." to "Category boundaries were selected based on our estimation of the utility of the lines and their anticipated performance for neuron segmentation."

Line 450. Changed "Categories 3 and 4…" sentence to "Categories 3 and 4 were divided based on estimated difficulty of manual segmentation combined with intuition about future segmentation algorithm improvements, such that Category 3 lines are expected to be tractable for segmentation, whereas Category 4 lines are more challenging."

Line 687 (Figure 1 Supplement 1). Changed "the full CNS expression pattern" to "2D MIPs of MCFO and full CNS expression patterns"

P7, l237: From the text, it was not clear to me if the alignment to JRC2018 included also the EM dataset. As this is essential for the whole matching process, it would be helpful to explicitly mention that alignment has been done for both, LM and EM data.

Line 237. Changed "samples" to "LM and EM data".

Comments on Described Search ApproachesP7, l250: The reference (Otsuna et al. 2022) seems to contain important details on preprocessing and one of the used matching algorithms. However, the paper is still unpublished and is marked in the reference list as "in preparation". For transparency and replicability reasons, the authors should ensure that algorithmic details are accessible to the public and either provide a prepublication highlighting the respective method or include method details in this paper.

Development of CDM and PPPM search algorithms and associated pre- and post-processing optimizations has proceeded in parallel with the MCFO data release and NeuronBridge application described in the paper. Mais et al., 2021 describes in detail their work to optimize PPPM. CDM improvements since Otsuna et al., 2018 will be described in Otsuna et al., 2023, which isn't ready yet. While we view the search approach evaluations as showing that neuron matches can be found with CDM and PPPM, the evaluation can't be comprehensive across all neurons, datasets, and algorithm variations.

P7, l244 + l255 and P15, l419: PPPM requires segmentation of neurons in LM images. However, information on how this segmentation is performed is missing. Please clarify.

Line 244. Added "(identified in our samples by PatchPerPix segmentation)" and another Hirsch 2020 citation.

P8, l269+Providing a custom processing and search capability for private 3D image stacks is certainly a great service. However, it might also raise concerns of potential users in terms of trust in respect to potential disclosure of private research data after upload. To be transparent in this respect, it would be beneficial to add information on data handling procedures in terms of privacy and security. Simply said, as a user I would like to know what happens with my data and its derivations after upload and processing? Is it deleted? Who has access to it?

This will be addressed on the revised website UI with the addition of an "Uploaded Data Usage and Retention Policy" which the user needs to agree with before uploading data. The full text of the policy is reproduced here:

"By uploading data to the NeuronBridge alignment and search service, you acknowledge that the data will be converted to a MIP of the aligned image. The image stacks, aligned MIP image, and any other derived data will only be accessible to you, under your personal login, and the HHMI Janelia developers for maintenance purposes until you delete the search. The data is not accessed by Janelia personnel for any other reason. You further acknowledge that this service is free and HHMI, its employees, and officers accept no liability for its use and do not guarantee or warrant the accuracy or utility of the output."

Comments on Search Approach Evaluation:The described evaluation approach is only partially feasible to support the very general claim of the authors that "NeuronBridge rapidly and effectively identifies neuron matches …." (P1, l28; and other parts of the manuscript): Although the evaluation appears to be straight forward and comprehensive at first glance, it lacks scientific rigor in several dimensions (see also comments below) and does not follow established evaluation standards in information retrieval. I understand that a full evaluation of the system according to the state of the art might be out of scope of this paper. However, in this case the claim should highlight that the performed experiments provide a demonstration of the potential of the method to effectively identify neuron matches, but not a proof.If the authors want to keep the generality in the claim, they should consider to revise this section in respect to the design of their experiments and the formal selection, justification and use of appropriate quality measures introduced by the information retrieval community. (The following manuscript might serve as an entry point and design further experiments: https://nlp.stanford.edu/IR-book/html/htmledition/evaluation-in-information-retrieval-1.html)Besides of the more high-level comment above, I have some remarks that might further illustrate my concerns:– The numbers of selected query items and performed queries are too low to draw general conclusions.– Transparency of the experience design:In both experiments sets of 10 and 9 neurons respectively were selected to perform query result analysis. It remains unclear based on what criteria exactly these neurons were selected, leaving open if there is any bias in the selection which would disqualify the results.

We acknowledge the limited set of neurons examined in the evaluation of CDM and PPPM search, and tried to weight the claims accordingly in lines 305 and 309 of the submission. We agree more examples could be useful, but providing them hasn't proven feasible during the revision period. As mentioned by the reviewer, a full evaluation based on the state of the art in the field of information retrieval is beyond the scope of the paper. We attempted to select a range of neurons for analysis that were within our domain of anatomical expertise, favoring accuracy of the evaluation over randomization of the analyzed neurons. We made the following modifications to further temper the breadth of claims:

Line 28. Added "We demonstrate the potential of NeuronBridge to…"

Line 278. Replaced "We evaluated…" with "We performed limited evaluations of…"

Line 285. Added ", and we restricted our analyzed set of neurons accordingly"

Forward, "qualitative", Analysis: A true qualitative evaluation would require the repetition of retrieval experiments with several experts and an investigation of the question if there is e.g. an inter-observer variability, which seems not to be the case here. In this context also information on the number and professional background of the persons who judged on the query results should be added.

Please see response under *Comments on Data Pre-Processing and Annotation.*